

# Alternative dynamic regimes of plankton communities in perturbed environments

Guido Occhipinti[1,2], Davide Valenti[3], and Paolo Lazzari[1,2]

[1]National Institute of Oceanography and Applied Geophysics - OGS, via Beirut 2, Trieste, I-34014, Italy
[2]NBFC, National Biodiversity Future Center, Palermo, I-90133, Italy
[3]Dipartimento di Fisica e Chimica "Emilio Segrè" - Università degli Studi di Palermo, Viale delle Scienze, Ed. 18, Palermo, I-90128, Italy

**Correspondence:** Guido Occhipinti (gocchipinti@ogs.it)

**Abstract.** The existence of alternative dynamic regimes or equilibria has been widely observed in the biosphere and the climate system. In order to assess the potential impacts of climate change and develop effective mitigation and adaptation strategies, a comprehensive knowledge of these alternative regimes is crucial. We studied marine biogeochemical cycles, which are fundamental for sustaining ocean life and for climate regulation, with a biogeochemical model used for operational purposes.

We investigated whether the perturbation of the environment (e.g. air temperature, wind velocity, nutrient input) to extreme values can push biogeochemical cycles into a different regime. We have established that this phenomenon exists and that the system commonly responds reversibly to the perturbation of the environment, i.e. when the perturbation is removed the original system regime is recovered. Depletion of nutrients and increase in wind velocity can induce hysteresis in the dynamic regimes associated with changes in the planktonic trophic web, which sustains the biogeochemical cycles. The large number of

numerical simulations under a vast range of environments and methodology, comprising demographic stochasticity, underpins the generality of the results and the sensitivity analysis of the model parameters confirms the accuracy of the model even under extreme environments. The occurrence of alternative dynamic regimes in a modern marine biogeochemical model, supported by field observations of regime shifts in plankton, suggests its use in predicting the state of the ocean under climate change.

## 1 Introduction

Alternative dynamic regimes occur when a specific pattern, persistent in time with the same characteristics, of the system under study can be observed with differences in the magnitude of its properties. A well-known example is coral reefs, which can switch from a regime with a coral-dominated reef to an algal-dominated reef (Mumby et al., 2007; Norström et al., 2009; Schmitt et al., 2019). The shift between two regimes can have profound ecological and social consequences, can be difficult to reverse and difficult to predict (Scheffer et al., 2001; Scheffer and Carpenter, 2003; Beisner et al., 2003; Schmitt et al.,

2019; Suding et al., 2004). A comprehensive knowledge of alternative dynamic regimes is required for ecosystem management and conservation, as they may constrain recovery actions (Carpenter et al., 1999; Conversi et al., 2015; Gladstone-Gallagher et al., 2024; Guarini and Coston-Guarini, 2024; Van De Leemput et al., 2016) and can cause (un)desired regime shifts (Stecher and Baumgärtner, 2022). Examples of alternative dynamic regimes have been found in aquatic and terrestrial ecosystems



(Donovan et al., 2018; Lindegren et al., 2012; Hewitt and Thrush, 2010; Petraitis, 2013; Schooler et al., 2011; Schröder et al.,
2005; Suding et al., 2004; Vandermeer et al., 2004) and physical systems (Van Westen et al., 2024; Wendt et al., 2024), but
marine biogeochemical cycles are still poorly investigated in this context, with a focus on plankton communities (Bode, 2024;
Kléparski et al., 2024; Molinero et al., 2013; Phlips et al., 2021; Soulié et al., 2022).

Biogeochemical cycles refer to the natural pathways through which essential elements and compounds circulate between
living organisms and the physical environment. These cycles, including the carbon, nitrogen and phosphorus cycles, are fun-
damental to the maintenance of life on Earth as they regulate the availability of nutrients necessary for biological processes
(Falkowski et al., 1998). Ocean biogeochemical cycles contribute to mitigating the climate crisis by absorbing carbon diox-
ide from the atmosphere (Falkowski et al., 1998). A fundamental role is played by plankton, which act as mediators of these
processes and are responsible for half of the Earth's oxygen production (Naselli-Flores and Padisák, 2023). Due to difficulties
in field observation in the marine environment, marine biogeochemical cycles are also studied using numerical models that
simulate interactions among plankton communities, microbial loops, and nutrient cycles (Fennel et al., 2022). These models
offer valuable insights into the state, variability, and long-term trends of marine ecosystems, supporting scientific research and
policy decisions. Their importance is underscored by their integration into major initiatives such as the European Copernicus
Marine Service (CMS) and the Coupled Model Intercomparison Project Phase 6 (CMIP6) (Eyring et al., 2016). Anthropogenic
and natural climate change has already led to a shift in the dynamic regimes of marine ecosystems (Knowlton, 1992; Hare and
Mantua, 2000; Möllmann and Diekmann, 2012), it is imperative to assess whether modern biogeochemical models are able to
predict alternative dynamic regimes and to identify which factors may cause a regime shift.

We assessed the occurrence of alternative dynamic regimes in the biogeochemical flux model (BFM) (Vichi et al., 2020),
which is used for operational purposes under the European initiative CMS (Salon et al., 2019). We investigated as dynamic
regimes the seasonal properties of the planktonic ecosystem (chlorophyll bloom and deep chlorophyll maximum) and biogeo-
chemical functions (e.g. total production of chlorophyll and nutrient recycling). A large range of intensities of environmental
forcings has been applied to push the system to alternative regimes. Since climate change is exacerbating extreme events (Gru-
ber et al., 2021), the model was run under extreme values for the external forcings to assess the impact on alternative regimes.
Particular attention has been given to wether the response of the system to the perturbation of the environmental conditions is
reversible or hysteresis occurs. The conservation of ecosystems need knowledge on hysteresis because it can limit the ability
to recover a desired state (Carpenter et al., 1999; Gladstone-Gallagher et al., 2024; Guarini and Coston-Guarini, 2024; Van
De Leemput et al., 2016). The occurrence of alternative dynamic regimes, reversibility and hysteresis was assessed using three
methods: i) a sequence of simulations with increasing and then decreasing external forcings; ii) an ensemble of simulations
with increasing external forcing, where an ensemble of initial conditions was run for each forcing value; iii) an ensemble of
simulations with increasing external forcing, with demographic stochasticity implemented in the model. Demographic stochas-
ticity origins from the uncertainty in birth and death events (Lindo et al., 2023; Melbourne, 2012) and may push a system to
alternative regimes (DeMalach et al., 2021).

This is the first time, to our knowledge, that alternative dynamic regimes have been comprehensively studied in biogeochem-
ical models. We have investigated more than 4000 possible scenarios, taking into account each dynamic regime, the magnitude





of the forcings and the numerical experiments, in order to obtain general results suggesting under which environmental forcing

alternative regimes exist and whether the system is reversible or hysteresis occurs. The ability to run simulations over a period of 100 years also confirmed that the regimes are not transient, which is a risk when short-term observations are used to identify alternative dynamic regimes (Schröder et al., 2005).

## 2 Methods

### 2.1 Biogeochemical model

Plankton dynamics and biogeochemical cycles in a 1-D water column are produced by the coupling between the Biogeochemical Flux Model (BFM) (Vichi et al., 2020) and the General Ocean Turbulence Model (GOTM) (Burchard and Petersen, 1999). This model configuration is described in detail in (Álvarez et al., 2023) and in the BFM code manual (Vichi et al., 2020). Here we give a brief overview of the model, focusing on the plankton dynamics, and a more detailed overview is provided in the supplementary material.

The biogeochemical flux model (BFM) consists of a system of $N_{tot} = 54$ ordinary differential equations describing the cycling of carbon, phosphorus, nitrogen, ammonium, silica and oxygen through inorganic, living, dissolved and particulate organic phases. The transmission of light in the water column is resolved in 33 wavebands centered on wavelengths ranging from $250$ to $3700\,nm$ (Lazzari et al., 2021b), with the light being absorbed and scattered by the water, phytoplankton, chromophoric dissolved organic matter (CDOM) and detritus (Álvarez et al., 2023). GOTM, which applies vertical mixing to the

BFM variables, can describe a one-dimensional water column representing saline, thermal and turbulence dynamics. BFM and GOTM are combined at runtime using the Framework for Aquatic Biogeochemical Models (FABM) (Bruggeman and Bolding, 2023). Since the computational resources required for a 1-D simulation are much smaller than for a global 3-D model, we were able to perform a large number of long simulations, but the horizontal gradients at the application sites must be either negligible or parameterized to properly describe reality.

The BFM describes the lower trophic level of marine ecosystems with primary producers (phytoplankton), predators (zooplankton) and decomposers (bacteria). These groups are then subdivided into plankton functional types (PFTs), which differ in their ecological function. Phytoplankton are subdivided into diatoms (P1), nanoflagellates (P2), picophytoplankton (P3) and dinoflagellates (P4). Zooplankton in nanoflagellates (Z6), microzooplankton (Z5), omnivorous mesozooplankton (Z4) and carnivorous mesozooplankton (Z3). Bacteria (B1) are represented by a single PFT and are responsible for the fundamental aspect

of recycling organic compounds into inorganic constituents such as nitrates, phosphates and silicates. The growth of primary producers is limited by nutrients (phosphorus and ammonium), light and temperature; diatoms also require silicate to form a protective shell. In the original version of the BFM, the rate of metabolic processes increases monotonically with the water temperature (Lazzari et al., 2021a). We have adopted the more realistic description of Blackford et al. (2004) so that the rate of these processes increases with temperature up to a threshold ($32°C$), after which the rate slowly decreases, representing

enzyme degradation at high temperatures (see supplementary material Eq.4). The grazing of phytoplankton by zooplankton is described by a Holling type II functional response (Gentleman et al., 2003). Both phytoplankton and zooplankton release $CO_2$



by respiration and organic matter by excretion. The main function of bacteria is the remineralization of particulate detritus and dissolved organic matter, but they can also act as competitors to phytoplankton by taking up inorganic nutrients from the water, depending on their internal nutrient quota.

The coupled GOTM-BFM model requires atmospheric and biogeochemical forcing. We have implemented the model at the site *Bouée pour l'acquisition de Séries Optiques à Long Term* (BOUSSOLE) with the 1-D water column setup described in (Álvarez et al., 2023). Atmospheric forcing is provided by the ECMWF ERA5 dataset, spectral light composition at the sea surface by the OASIM model (Lazzari et al., 2021b) and temperature, salinity and biogeochemical variables come from the reanalysis of the biogeochemistry of the Mediterranean Sea (Cossarini et al., 2021). The model was numerically integrated

with climatological-averaged forcings from 2000 to 2050 with a time step of $10min$, and the day-to-day averages of the last 5 years of the model output were stored.

## 2.2 Alternative dynamic regimes

We are interested in the existence of alternative stable states in our modeled ecosystems. Since real ecosystems are never stationary because natural populations always fluctuate (e.g. due to a seasonal environment), we adopt the nomenclature of

Scheffer and Carpenter (2003) and call the stable states dynamic regimes. A dynamic regime is a specific pattern of the system that persists in time with the same characteristics. Two types of alternative dynamic regimes are possible (Beisner et al., 2003): i) different regimes exist simultaneously under the same conditions (parameters), and the ecosystem can change from one regime to another due to disturbances that directly affect the state variables; ii) different regimes exist under different conditions, e.g. due to a change in environmental forcings that alter the ecosystem regime.

When a parameter changes due to a perturbation, the dynamic regime of the ecosystem may change, tracing a path through the space of possible dynamic regimes. When the perturbation is relaxed and the parameter returns to its original value, two scenarios are possible. The dynamic regime can return to its original value by following the same trajectory it followed under the perturbation, in which case the system is reversible. If the regime moves along a different trajectory, the system exhibits hysteresis (Beisner et al., 2003), which can only occur if alternative dynamic regimes exist for the same value of the parameter.

After hysteresis, it may be more difficult or even impossible for the system to return to its original state, which is why predicting hysteresis is fundamental for ecologists and managers.

A minimal model showing alternative dynamic regimes and hysteresis is the Hill model (Scheffer et al., 2001), which is a mathematical model describing an ecosystem property $X$ under environmental forcings:

$$\frac{dX}{dt} = a - bX + rf(X) \tag{1}$$

where $a$ is an environmental condition that promotes $X$, $b$ is the rate of decay of $X$, $r$ is the rate at which $x$ recovers, where

$$f(X) = \frac{X^p}{X^p + h^p} \tag{2}$$

$f(X)$ is a function with a threshold, called the Hill function, and $p$ a parameter determining the steepness of the switch occurring around $h$. For phytoplankton, one could interpret $X$ as biomass, $a$ as nutrient input, $b$ as predation and $r$ as migration.





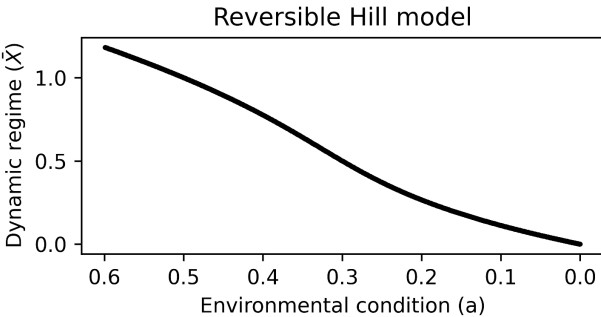 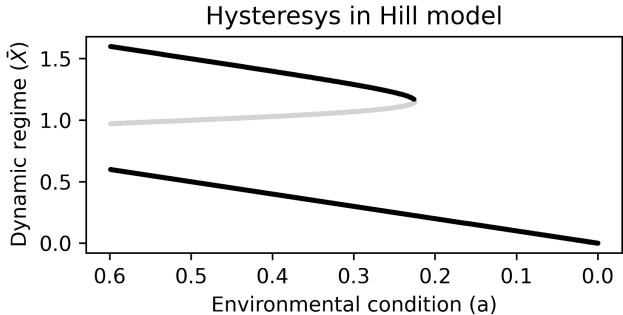

**Figure 1.** Bifurcation diagrams showing the response of the Hill model's dynamic regimes to the environmental condition $a$. The left panel show a reversible response of the Hill model with parameters $p = 2$ and $b = r = h = 1$. The right panel show the hysteresis in the Hill model with parameters $p = 18$ and $b = r = h = 1$. Black (grey) lines are the stable (unstable) dynamic regimes of the Hill model.

We are interested in the steady state value $\bar{X}$ of the state variable $X$, or the dynamic regime as in the BFM case, and in particular to its dependency on the model parameters $p$ and $a$. For different values of $p$, $\bar{X}$ responds differently to perturbation of the environmental condition $a$. For example for $p = 2$ (and $b = r = h = 1$), for each value of $a$ there is only a corresponding $\bar{X}$, showing therefore a reversible response to the variation of $a$ (see left panel of Fig.1). For higher $p$, two values of $\bar{X}$ are possible for the same $a$, giving rise to hysteresis (see right panel of Fig.1).

### 2.3 Experiments setup

To investigate the possibility of alternative dynamic regimes in the modeled ecosystem, we identified a list of target indicators focusing on the plankton community and biogeochemical cycles. We examined 6 indicators related to the chlorophyll bloom, 6 indicators related to the deep chlorophyll maximum (DCM), and 4 indicators related to the aggregated system production and cycling. The full list of target indicators with the computation method is given in Tab.2.

We investigated the occurrence of alternative dynamic regimes under the perturbations of 5 parameters, analogues to the parameter $a$ in the Hill model, of the coupled GOTM-BFM model related to the environment in which the ecosystem is located, thus we refer to them as environmental forcings. The atmospheric forcings have a temporal dimension with seasonal variability and no inter-annual variability, the marine forcings also depend on the depth. A constant value has been added to the forcings, changing their annual mean, to assess the possibility of alternative regimes under extreme environmental conditions. The full list of studied forcings and their range of variability is given in Tab.1.

We performed 4 experiments consisting of a set of simulations, with a different a protocol to reach the dynamic regime. This is done to understand if the results depend on the specific choices used to construct the experiment. For each experiment we considered all the perturbations of the 5 environmental forcings. A schematic of the simulation procedure, for each experiment, is shown in Fig.2. Prior to the experiments, a spin-up simulation with a duration of 40 years was carried out to enable a restart in which the model had already relaxed to the dynamic regimes.





**Table 1.** List of the forcings. In the column "Variation of the mean" the sign "+" indicates that the mean is increased of the range of values between square brackets, while the sign "-" indicates that the mean is decreased. The range of values of the original means for some forcings is due to a vertical variability through the water column.

| Forcing | Variation of the mean | Original mean |
|---|---|---|
| 10m wind (x) | + [0-15] | 0 $[m/s]$ |
| 2m air temperature | + [0-15] | 17 $[^{\circ}C]$ |
| 2m air temperature | - [0-15] | 17 $[^{\circ}C]$ |
| suspended particulate matter | + [0-1.2] | 0.1 $[g/m^3]$ |
| phosphate nudging | - [0-0.25] | [0.08-0.9] $[mmol\,P/m^3]$ |
| oxygen nudging | - [0-100] | [200-250] $[mmol\,O/m^3]$ |

Table 2: List of the target indicators of the ecosystem studied as system response to the perturbation of parameters, divided in three major categories: CHL bloom, DCM, aggregated ecosystem production and cycling.

| Indicator | Description | Method of computation |
|---|---|---|
| Max CHL | maximum value of CHL through the year | the first 10 meters in depth are summed and then the maximum value of CHL during the year is taken |
| Time of Max CHL | time of the year when the maximum of CHL is reached | number of days since the beginning of the year when the maximum of CHL occurs |
| Duration of Max CHL | temporal length of the CHL bloom | it is the full width at half maximum of the CHL bloom |
| Biomass ratios | distribution of biomass between bacteria, phytoplankton and zooplankton during CHL bloom | the carbon concentration of each group is computed as the temporal average over one month centered on the "Time of Max CHL", averaged over the first 10m in depth and normalize over the total carbon concentration of all groups |
| P Biomass ratios | distribution of biomass phytoplankton functional types during CHL bloom | the carbon concentration of each phytoplankton functional type is computed as the temporal average over one month centered on the "Time of Max CHL", averaged over the first 10m in depth and normalized over the total carbon concentration of all phytoplankton |



| Z Biomass ratios | distribution of biomass zooplankton functional types during CHL bloom | the carbon concentration of each zooplankton functional type is computed as the temporal average over one month centered on the "Time of Max CHL", average over the first 10m in depth and normalized over the total carbon concentration of all zooplankton |
|---|---|---|
| Intensity of DCM | maximum value of CHL below surface waters | the total CHL is temporal averaged over the months June, July, August and the maximum is found through the vertical distribution |
| Depth of DCM | depth at which is found the maximum value of CHL below surface waters | depth at which is found the "Intensity of DCM" |
| Width of DCM | vertical distribution of the high CHL waters | it is the full width at half maximum computed over the temporal averaged CHL |
| Biomass ratios | distribution of biomass between bacteria, phytoplankton and zooplankton in the DCM | the carbon concentration of each group is computed as the temporal average over June, July August, computed at the "Depth of DCM" and normalized over the total carbon concentration of all groups |
| P Biomass ratios | distribution of biomass phytoplankton functional types in the DCM | the carbon concentration of each phytoplankton functional type is computed as the temporal average over June, July August, computed at the "Depth of DCM" and normalized over the total carbon concentration of all phytoplankton |
| Z Biomass ratios | distribution of biomass zooplankton functional types in the DCM | the carbon concentration of each zooplankton functional type is computed as the temporal average over June, July August, computed at the "Depth of DCM" and normalized over the total carbon concentration of all zooplankton |
| Total CHL (0-100m) | CHL produced in the first 100m during 1 year | the CHL is summed over the whole year and through the first 100m in depth |
| Mean O2 (0-200m) | Mean oxygen produced in the first 200m during the year | the dissolved oxygen is averaged over the whole year and through the first 200m in depth |





| Total N remin. (0-100m) | Ammonium remineralized by bacteria and zooplankton in the first 100m during 1 year, associated to nutrient recycling | the N remin. is summed over the whole year, through the first 100m in depth and through the ratio remineralized by each species involved in the process |
|---|---|---|
| Total P remin. (0-100m) | Phosphate remineralized by bacteria and zooplankton in the first 100m during 1 year, associated to nutrient recycling | the P remin. is summed over the whole year, through the first 100m in depth and through the ratio remineralized by each species involved in the process |

### 2.3.1 EXP-SEQ: Sequential simulations

The first experiment (EXP-SEQ) deals with the question of whether the dynamic regimes under the variation of an environmental forcing follow the same trajectories when the forcing intensity is increased or decreased. Two sets of 15 sequential simulations with a length of 45 years were carried out for each forcing: in the first set of simulations, i.e. forward simulations, the mean value of the forcing was increased (decreased); the second set, i.e. backward simulations, restarts from the end of the first set and the mean value is decreased (increased), in this set the concentration of living species is perturbed at the beginning of the simulation. For each intra-cellular quota element the perturbation is $+10^{-2}C$ for carbon , $+10^{-4}N$ for nitrogen and $+10^{-5}P$ for phosphorus, equally for each species. This perturbation prevents extinction and allows observe new alternative equilibria. This perturbation represents a demographic variability, always present in natural populations which is not described by the model and can prevent extinction. The day-to-day mean of the last 5 years of each simulation is stored. The dynamic regimes are computed for each simulation and plotted in a bifurcation diagram to determine whether the system is reversible or hysteresis occurs. The sequence of simulations is repeated separately for each forcing.

### 2.3.2 EXP-INI: Random initial condition simulations

The second experiment (EXP-INI) deals with the question of whether the model can relax to alternative dynamic regimes under different initial conditions and whether this behavior is related to the magnitude of the environmental forcings. For each magnitude of the investigated forcing, e.g. for a mean 10m wind (wind at 10 meters above the sea surface), we performed an ensemble of 50 simulations with different initial conditions: i) for carbon and nutrients of the plankton species, a random value ranging $\pm70\%$ from the restart value provided by the spinup simulation with climatological forcings; ii) for the remaining variables, the restart of the spinup simulation. The simulation ensembles are then used to generate bifurcation diagrams, which allows to observe whether alternative dynamical regimes exist simultaneously for the same magnitude of environmental forcings. The simulations ensemble is repeated separately for each forcing.



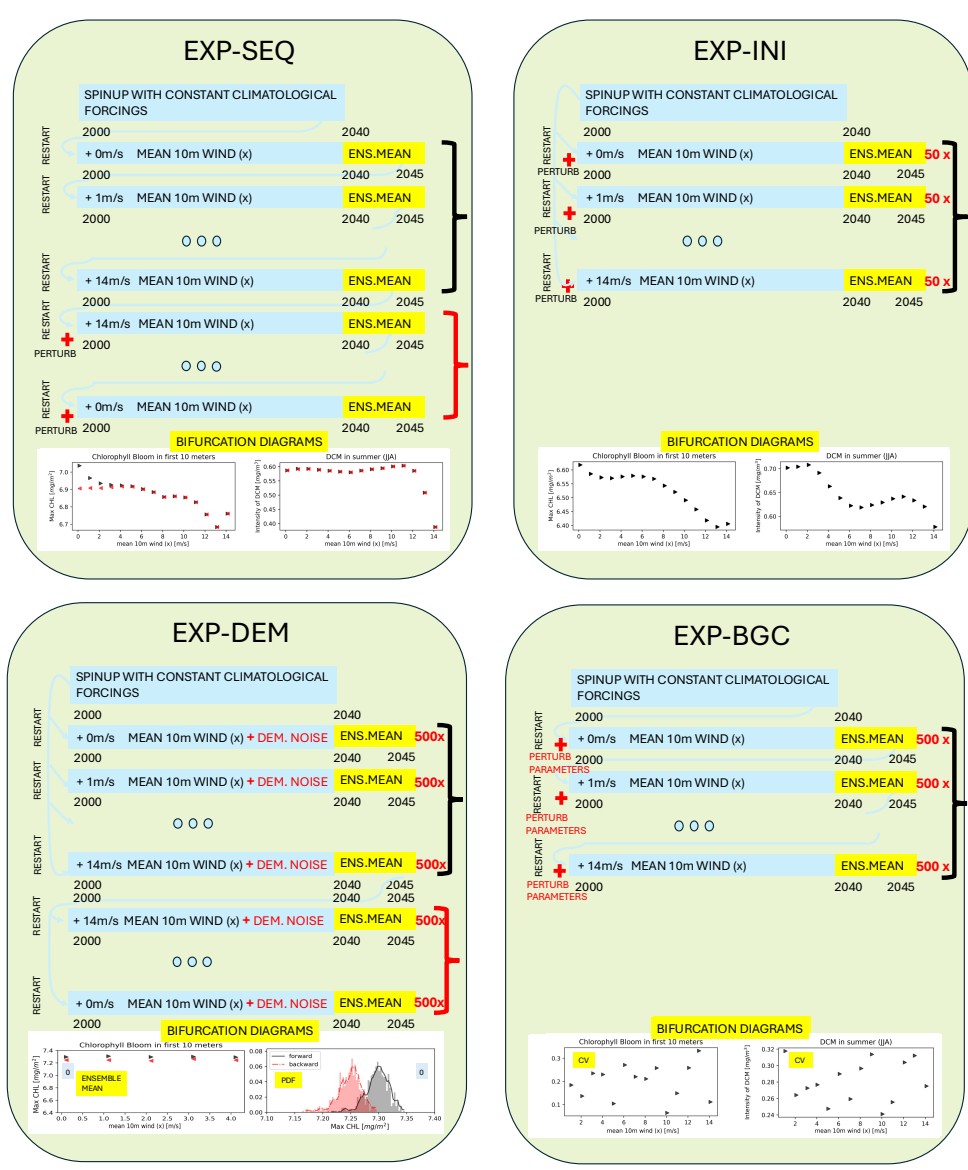

**Figure 2.** Protocols of the simulation experiments EXP-SEQ, EXP-INI, EXP-DEM, and EXP-BGC. The experiments methodology is described in Sect.2.3. For each EXP we considered the effects of the perturbation of the 6 environmental forcings, listed in Tab.1, over all the target indicators, listed in Tab.2. Here the forcing 10m wind (x) is taken as an example.





### 2.3.3 EXP-DEM: Demographic noise simulations

The third experiment (EXP-DEM) deals with the role of stochasticity in pushing the system from one dynamic regime to another, and with checking whether this behavior is related to the magnitude of the environmental forcings. If there are alternative dynamic regimes, a stochastic perturbation can push the system into the region of attraction of another dynamic regime. In this experiment, we incorporated the stochasticity of birth and death events, which are assumed to be independent between individuals, like the demographic noise in the biogeochemical model. This effect can be represented by a multiplicative white noise acting on the biomass of the plankton species with a sublinear scaling with respect to the biomass (Arnoldi et al., 2019). In the BFM, we applied the noise to the biomasses of bacteria, phytoplankton and zooplankton. The biomass $X^i$ of a general PFT is evolved by the following stochastic differential equation

$$dX^i = BFM_{X^i}(\boldsymbol{X}, t)dt + D^i k^i \sqrt{X^i} dW, \tag{3}$$

where $BFM_{X^i}$ denotes the deterministic dynamics of the biomass $X^i$ provided by the deterministic version of the BFM, $\boldsymbol{X}$ is the vector of the $N_{tot}$ state variables, $W$ is a Wiener process and $k^i = 1\sqrt{mgC/m^3}$ is a normalization factor. We studied three noise intensities $D^i = [0.01, 0.001, 0.0001]d^{-1/2}$, equal for each PFT. For each magnitude of the investigated forcing we performed an ensemble of 500 stochastic simulations from which we computed an averaged dynamic regime, used to generate bifurcation diagrams, and the probability distribution function (PDF) of the regime. The simulation ensemble is repeated separately for each forcing. We performed two set of simulations one using as initial conditions the restart of the spinup simulation (forward simulations), another using as initial condition the ensemble mean of the restarts of the forward simulation with the largest environmental forcing (backward simulations). These two sets are compared similarly as done in the experiment EXP-SEQ.

### 2.3.4 EXP-BGC: Biogeochemical parameters sensitivity analysis

The forth experiment (EXP-BGC) deals with the variability of the dynamic regimes under the perturbation of 10 parameters of the biogeochemical model (BFM), that characterize the plankton dynamics. For the 4 phytoplankton PFTs, we perturbed 2 parameters regulating primary productivity: the maximum specific productivity ($p\_sum \in [1.25, 4.75]d^{-1}$) and the reference ratio $Chl:C$ ($p\_qlcPPY \in [0.005, 0.038]mgChl/mgC$). For the microzooplankton, characterizing the plankton dynamics being in the center of the modeled food web (Occhipinti et al., 2023), we perturbed two parameters regulating its growth: the assimilation efficiency ($p\_pu \in [0.35, 0.65]$) and the fraction of activity excretion ($p\_pu\_ea \in [0.35, 0.65]$). For each magnitude of the forcing studied we performed an ensemble of $N_{ens} = 500$ simulations with different parameters. The simulation ensemble is repeated separately for each forcing. The simulation ensembles are then used to compute the coefficient of variation (CV) of the dynamic regimes at each magnitude of the environmental forcings, with the aim of investigating a relationship with the parameter uncertainty and the strength of the forcings. The CV is computed for a dynamic regime $\chi$ over the simulation



ensemble as

$$CV_\chi = \frac{\mu_\chi}{\sigma_\chi} = \frac{\sum_{\alpha=1}^{N_{ens}} \chi_\alpha/N_{ens}}{\sqrt{\sum_{\alpha=1}^{N_{ens}} (\chi_\alpha - \mu_\chi)^2/N_{ens}}}, \tag{4}$$

where $\mu_\chi$ and $\sigma_\chi$ are, respectively, the mean and standard deviation of the dynamic regime $\chi$ over the ensemble of simulations, numbered $\alpha = 1, ..., N_{ens}$, and $\chi_\alpha$ is the dynamic regime computed for the simulation $\alpha$.

### 2.4 Nudging of biogeochemical variables

Two of the considered environmental forcings in Tab.1 are not forcings external to the biogeochemical model but nudging of biogeochemical variables. Therefore, some clarification on the phosphate and oxygen nudging forcings are needed (see Tab.1). Phosphate nudging means that the variable phosphate relaxes during the simulation to a certain vertical profile with a certain relaxation time $\tau$, i.e. at each point of the vertical grid the phosphate variable is evolved as follows

$$\frac{\partial P}{\partial t}(z,t) = \left( \begin{matrix} \text{physical} \\ \text{processes} \end{matrix} \right) + \left( \begin{matrix} \text{biogeochemical} \\ \text{processes} \end{matrix} \right) - \left( \frac{P(z,t) - P_{nudging}(z,t)}{\tau} \right). \tag{5}$$

Therefore, our forcing is based on the increase or decrease of the values of the vertical profile to which the model relaxes ($P_{nudging}$). We have chosen to change the profile uniformly along the vertical grid, and we have chosen as relaxation time the value $\tau = 1\,year$ as in (Álvarez et al., 2023). Similar considerations apply to the oxygen nudging. Perturbing phosphate parameter is analogous to perturb the parameter $a$ in the Hill model.

## 3 Results

### 3.1 EXP-SEQ

In the experiment performing sequential simulations under variation of environmental forcing (EXP-SEQ), we found both reversibility and hysteretic response of the system.

Reversible response is found under the variation of air temperature (airt), suspended particulate matter (spm) and oxygen (oxy), not shown, the model visits alternative dynamic regimes (among those of Tab.2). The visited regimes are the same in the forward path, where the forcing increases, and in the backward path, where the forcing decreases (see Fig.3). The system is, thus, reversible under these forcings because the dynamic regime can return to its original value by following the same trajectory it followed under the perturbation.

The model showed the presence of hysteresis under the variation of wind velocity and phosphate nudging. The system never returns to the dynamic regime it was in before the perturbation of the environmental forcings. The results, confirmed by additional simulations over a period of 100 years, indicate therefore that it is not a transitory phenomenon.

In the supplementary material, Figures S1-S8, we show the response of all the target indicators presented in Tab.2. Oxygen nudging has no strong difference in dynamic regimes and the decrease in air temperature has an effect similar to the increase in the wind velocity, so that these two forcings are not shown. The hysteresis under the perturbation of phosphate nudging



**Figure 3.** Bifurcation diagrams showing the response of the model's dynamic regimes to environmental forcings in the EXP-SEQ setup. The left panels show a reversible response, the right panels show the hysteresis. The first line is a simplified representation of the response to environmental forcing using the Hill model (Scheffer et al., 2001), where $a$ is a parameter of the model. Black (gray) lines are the stable (unstable) dynamic regimes of the Hill model. Black (red) triangles are the solution of forward (backward) simulations. The methodology is presented in Sect.2.3.1. The second and third lines show the response of the dynamic regimes to the variation of 4 environmental forcings.

has been found to be related to a change in the composition of the trophic web. In the backward trajectory fewer species are present in the community, in particular bacteria, one phytoplankton population (picophytoplankton) and two zooplankton populations (het. nanoflagellates and microzooplankton). In contrast, hysteresis is not associated with a change in the trophic web under wind perturbation. The perturbation of an environmental forcing may lead to hysteresis in one target indicator, and to a reversible behavior in another. For example, the intensity of the DCM is reversible under wind perturbations. Furthermore,

we found that the decrease in air temperature has an effect similar to the increase in wind on the indicators, which is related to an increase in the mixing of the water column at lower temperature or stronger wind.



## 3.2 EXP-INI

Carrying out numerical simulations with random initial conditions on PFTs, under variation of environmental forcing (EXP-INI), we found that the system has, in the majority of the cases, reversible response (see Fig.4). Only under the forcing of
phosphate nudging, the variability in the PFTs initial conditions is able to push the system in alternative dynamics regimes for the same value of the forcing. However, these regimes show very little difference. Oxygen nudging has no strong difference in dynamic regimes and the decrease in air temperature has an effect similar to the increase in wind velocity, so that these two forcings are not showed.

The difference between the results of EXP-SEQ and EXP-INI is related to the fact that in EXP-SEQ we used as initial
conditions the end of the previous simulation in the sequence of simulations, e.g. the initial conditions of the forward simulation with mean wind $5\,m/s$ were the end of the simulation with mean wind $4\,m/s$. In the EXP-INI experiment, only the initial conditions for carbon, phosphorus and phosphate of the plankton functional types are perturbed. While in EXP-SEQ hysteresis was associated with strongly different alternative regimes in how the biomass is subdivided between plankton types, in EXP-INI no alternative regimes are found for the same magnitude of environmental forcing, not even in biomass ratios between
PFTs (compare Fig.S5 and Fig.S9, bottom three panels of both columns, see supplementary material).

## 3.3 EXP-DEM

In the experiment addressing the effect of demographic stochasticity, performing sequential simulations under variation of environmental forcing (EXP-DEM), the system presented generally a reversible response. As an example, we carried out the simulations under the variation of wind forcing (see Fig.5), in the range of values which showed hysteresis in EXP-SEQ.
Demographic stochasticity reduces the variability of dynamic regimes in response to the perturbation of the wind forcing, and the magnitude of the regimes changes less than in the deterministic experiment (EXP-SEQ), e.g. compare the first column panels in Fig.5. The ensemble mean of the dynamic regimes is different respect to their deterministic magnitude, indicating that demographic stochasticity changes the space of possible dynamic regimes. Hysteresis is found in the intensity of the chlorophyll bloom for any noise intensity, but not for $D = 0.001\,d^{-1/2}$ where the system is reversible, indicating a non-
monotonic relationship between noise intensity and hysteresis.

## 3.4 EXP-BGC

The dynamic regimes of the target indicators show variability in response to the perturbation of the configuration of the biogeochemical model parameters (the methodology is presented in Sect.2.3.4), as shown in Fig.6. The coefficient of variation (CV) of the dynamic regimes associated with seasonal patterns, chlorophyll bloom and deep chlorophyll maximum (DCM),
has a CV of the order of $10^{-1}$. Dynamic regimes associated with aggregated variables, e.g. total annual chlorophyll production (Total CHL), show less variability in response to parameter perturbation, with a CV of the order of $10^{-4}$.

The CV of the dynamic regimes does not show an increasing (or decreasing) trend related to the increase in wind velocity (see Fig.6). The CV under the perturbation of the biogeochemical model parameters represents the sensitivity of the model.

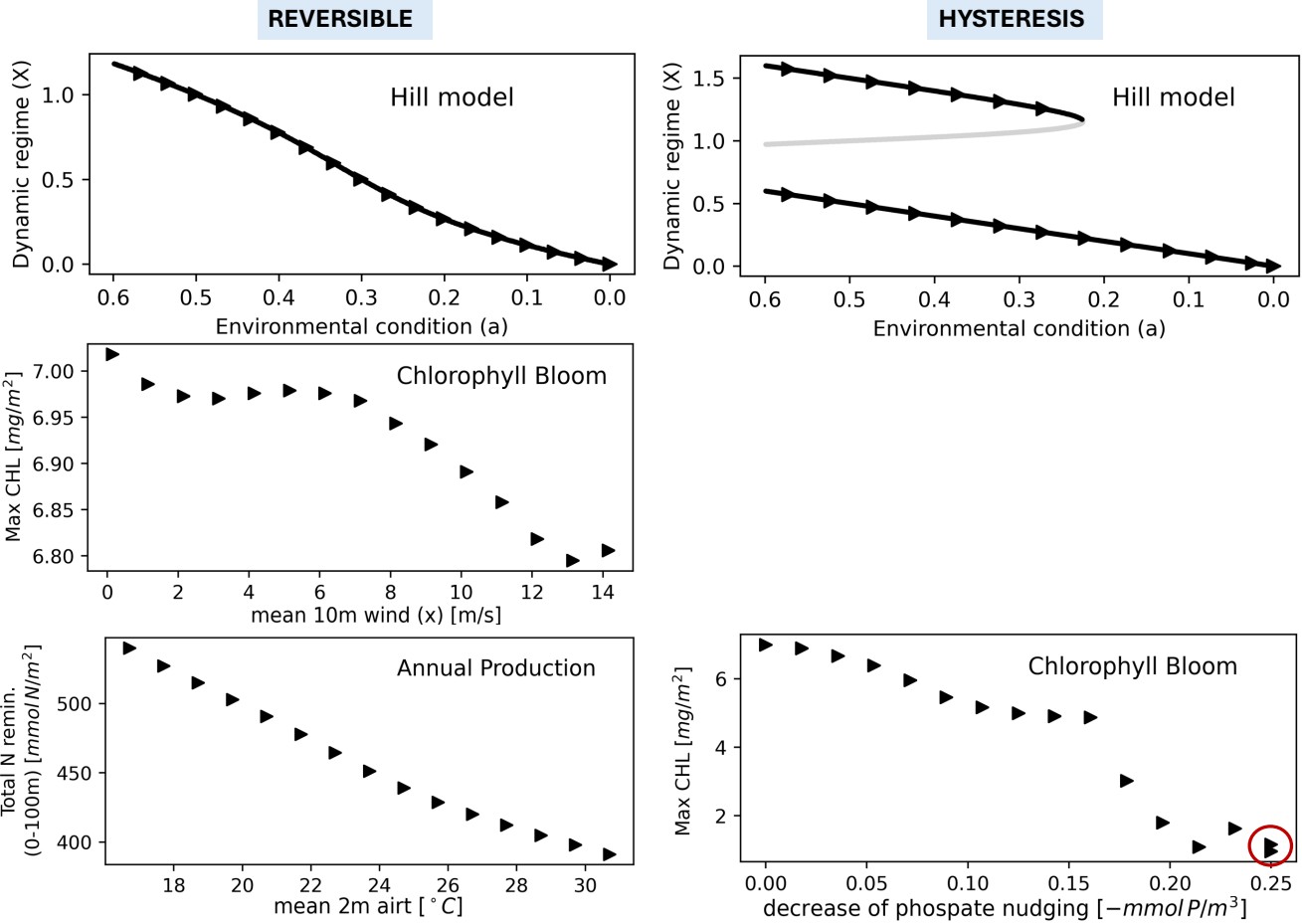

**Figure 4.** Bifurcation diagrams showing the response of the model's dynamic regimes to environmental forcings in the EXP-INI setup. The left panels show a reversible response, the right panels show the hysteresis. The first line is a simplified representation of the response to environmental forcing using the Hill model (Scheffer et al., 2001), where $a$ is a parameter of the model. Black (grey) lines are the stable (unstable) dynamic regimes of the Hill model. Each solution of the ensemble with random initial condition is showed by a black triangle. The methodology is presented in Sect.2.3.2. The second and third lines show the response of the dynamic regimes to the variation of 3 environmental forcings. Hysteresis occurs only for a strong decrease in the phosphate nudging and is highlighted by a red circle.

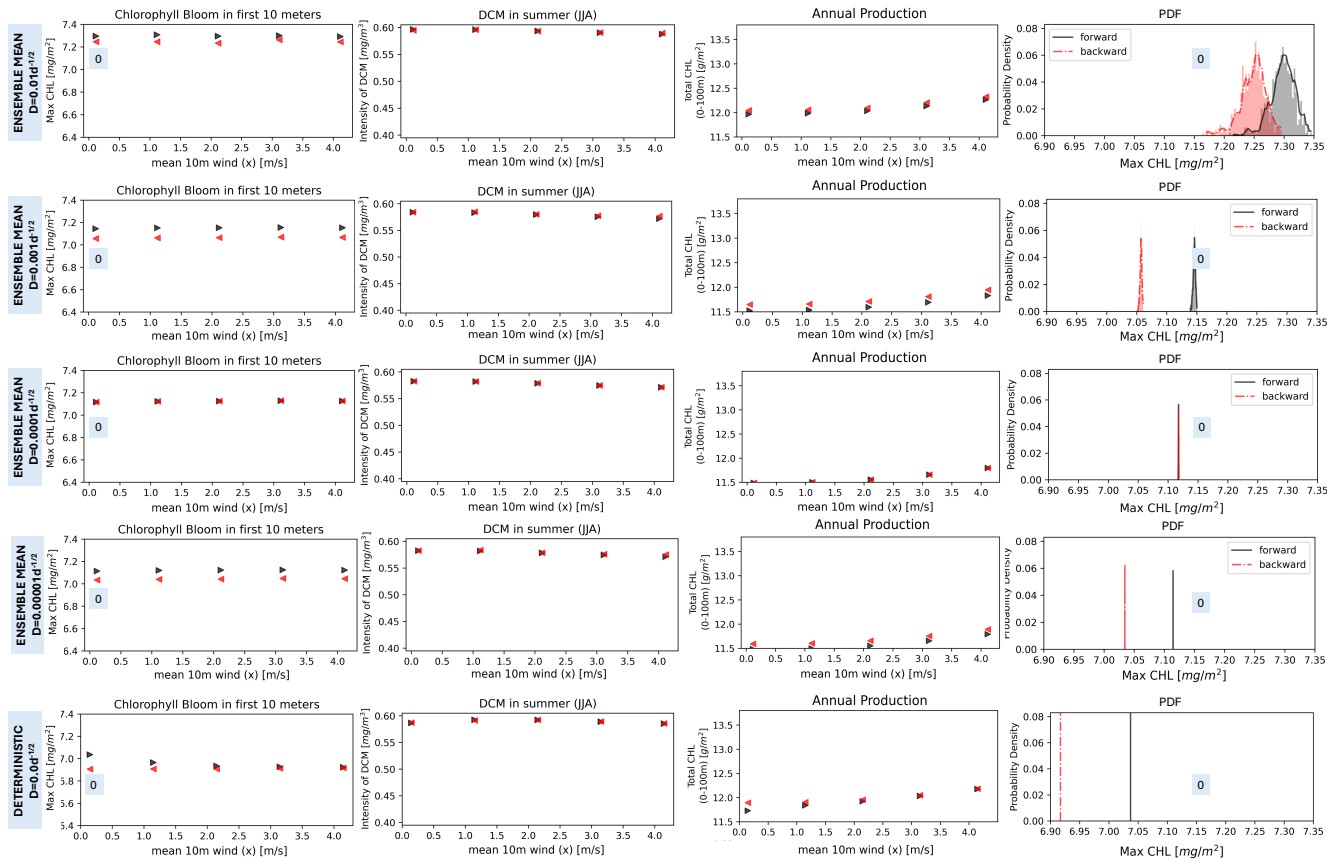

**Figure 5.** Bifurcation diagrams showing the response of the model's dynamic regimes to the increase of wind forcing in the EXP-DEM experimental setup. Each row shows the dynamic regimes for a different noise intensity decreasing from the top row ($D = 0.01\,d^{-1/2}$) to the bottom row (deterministic solution). Along a row the first three panels show the ensemble mean of a dynamic regime for the forward (black) and backward (red) simulations. The last panel, along a row, shows the probability density function of a dynamic regime (Chlorophyll bloom Max CHL) computed over the ensemble of simulations for the first value of wind forcing (unperturbed). The methodology is presented in Sect.2.3.3.





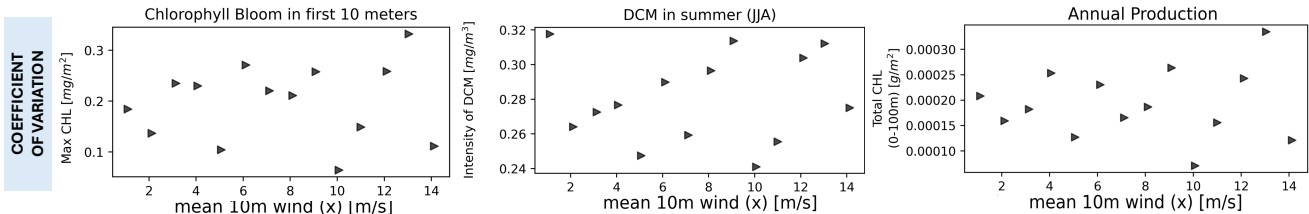

**Figure 6.** Bifurcation diagrams showing the response of the model's dynamic regimes to the perturbation of the model parameters under the increase in wind forcing (EXP-BGC, presented in Sect.2.3.4). The y-axis shows the coefficient of variation (CV) of the dynamic regimes over the ensemble of simulations with different parameters. The CV of the annual production dynamics regimes (rightmost panel) is 3 orders of magnitude lower than that of the seasonal dynamic regimes (first two panels).

Therefore, the sensitivity is not affected by extreme environmental conditions. In Fig.6 the sensitivity of the model (the dynamic
regimes) under the wind forcing is shown, but similar results (no trend in CV) are obtained with the other forcings and are therefore not shown.

## 4 Discussion

This study uses a set of numerical experiments to investigate how environmental conditions affect the possibility of alternative dynamic regimes in plankton communities and biogeochemical cycles. The large number of numerical simulations shows the
occurrence of alternative dynamic regimes under different environmental conditions, for most of the environmental forcings, and under the same environmental conditions when phosphate levels and wind velocity are perturbed. The latter scenario is related to the hysteresis phenomena.

 The deterministic experiments (EXP-SEQ, EXP-INI) show that the biogeochemical model possesses alternative dynamic regimes under different environmental conditions. The target indicators identify recurrent properties of marine plankton com-
munities (chlorophyll bloom, deep chlorophyll maximum, annual chlorophyll production) and biogeochemical processes mediated by plankton (oxygen production, nutrient remineralization). It was found that these indicators are related to climate variability (Molinero et al., 2013), so that alternative dynamic regimes are to be expected under different environmental conditions. The numerical identification of alternative regimes is supported by the observation of regime shifts (the change from one regime to another) in marine plankton systems (Aronés et al., 2019; Beaugrand, 2015; Bode, 2024; Hare and Mantua, 2000;
Kang et al., 2012; Molinero et al., 2013; Phlips et al., 2021; Soulié et al., 2022; Reid et al., 2016). Regime shifts are observed in all oceans when the time series span a decade or more (Bode, 2024). To our knowledge, this study is the first which investigates the occurrence of alternative dynamic regimes (and thus the possibility of regime shifts) in realistic biogeochemical models used with an operational scope, e.g. the BFM generates forecasts for the European Union under the Copernicus Marine Service (Salon et al., 2019). Previous work showed the possibility of alternative regimes in a mathematical model describing phyto-
plankton growth in a water column related to the strength of water mixing (Ryabov et al., 2010). The ability to predict regime





shifts may prove particularly important when forecasting state of the ocean under climate change scenarios (e.g. in Reale et al. (2022)).

In the extensive literature dedicated to understanding the drivers of regime shifts, sea surface temperature has been shown to be the primary driver of alternative regimes in marine plankton systems (Beaugrand, 2015; Bode, 2024; Chiba et al., 2012;

Morse et al., 2017; Reid et al., 2016; Soulié et al., 2022). In our modeled ecosystem, the increase in air temperature, which directly affects sea surface temperature, leads to a decrease in the intensity and duration of the spring chlorophyll bloom and, during the bloom, phytoplankton is no longer the most abundant group, since bacteria and zooplankton appear with comparable biomasses (see supplementary materials, Fig.S3 left panels). The increase of temperature leads to a deeper chlorophyll maximum in summer (see supplementary materials, Fig.S3 right panels). A decrease in all annual ecosystem functions (chloro-

phyll and oxygen production, phosphate and ammonium remineralization) is observed (see supplementary materials, Fig.S4). Stratification and wind may also play a role in pushing plankton into alternative dynamic regimes (Chiba et al., 2012; Peperzak and Witte, 2019; Ryabov et al., 2010). We found that an increase in wind is associated with a decrease in the intensity of the spring chlorophyll bloom but with an increase in its duration, and the greatest wind strengths studied reduced the intensity of the deep chlorophyll maximum (see supplementary materials, Fig.S1). An increase in all annual ecosystem indicators is ob-

served (see supplementary materials, Fig.S2), which is probably due to higher mixing of the water and higher nutrient content in the upper water column. Nutrient inputs are a key factor in coastal waters, in particular their decrease (Boersma et al., 2015; Feuilloley et al., 2020; Peperzak and Witte, 2019). We studied the effects of reducing phosphate nudging, which means that the phosphate in the water column relaxes to a lower concentration (see Sect.2.4). This could indicate an influx of oligotrophic water that reduces nutrient concentrations in the study area compared to concentrations prior to the perturbation of the nudging.

The decrease in phosphate nudging leads to a decrease in the intensity of the chlorophyll bloom and to a reorganization of the trophic web during the bloom (see supplementary materials, Fig.S5 left panels). The abundance of bacteria and zooplankton decreases drastically and phytoplankton dominates the trophic web. Diversity within the phytoplankton and zooplankton also decreases, with only picophytoplankton and heterotrophic nanoflagellates surviving because of the strongest decrease in phosphate input. Similar effects are observed for the deep chlorophyll maximum (see supplementary materials, Fig.S5 right panels),

and furthermore, it may be completely absent for some values of phosphate nudging (grey area in the figure). Annual chlorophyll production and nutrient remineralization decrease with a reduction in phosphate nudging (see supplementary materials, Fig.S6). The alternative dynamic regimes induced by phosphate nudging are the most different between the studied forcings in the modeled ecosystem, suggesting a leading role of nutrients in determining the dynamic regime visited by the ecosystem.

In addition to temperature, wind and nutrients factors, our results show that the increase in suspended particulate matter

(SPM) induces alternative dynamic regimes (see Fig.3). SPM has already been found to induce alternative regimes when studying the effects of sediment loading in coastal waters by retreating Antarctic glaciers (Neder et al., 2022; Sahade et al., 2015). Climate change is increasing SPM in coastal water causing a reduction in light availability, called coastal darkening, and inducing shifts in marine ecosystems (Aksnes et al., 2009). SPM influences the depth distribution of plankton due to light limitations (Neder et al., 2022), as shown for the depth of the summer deep chlorophyll maximum in Fig.3. Higher SPM leads

to a shallower chlorophyll maximum, which is associated with an increase in the concentration of phytoplankton relative to





zooplankton and bacteria (see supplementary materials, Fig.S7 right panels). We found that an increase in SPM is associated with an increase in the intensity of the chlorophyll bloom, but with a decrease in its duration and in the diversity (number of species) of phytoplankton and zooplankton (see supplementary materials, Fig.S7 left panels). A decrease in all annual ecosystem functions is observed (see supplementary materials, Fig.S8).

In the EXP-SEQ experiment, we observed a hysteretic response of the dynamic regimes to wind and phosphate environmental conditions (see Fig.3, right panels). The decrease in air temperature has a similar effect to wind increase and was therefore not shown. The possibility of the occurrence of hysteresis in plankton was confirmed by an experimental study on cyanobacteria under perturbed light conditions (Faassen et al., 2015). Although it is possible to confuse hysteresis with a transitory regime in natural ecosystems (Schröder et al., 2005), several studies have identified a hysteretic response in a wide range of systems,

from penguins to dune slack plants abundance (Bestelmeyer et al., 2011; Northrop et al., 2021; Litzow and Hunsicker, 2016; Schröder et al., 2005). We have performed additional simulations over 100 years (here not shown) to exclude the possibility that the observed hysteresis are only transitory. Feedback mechanisms are a possible cause of hysteresis (Holling, 1973; Kéfi et al., 2016; May, 1977; Scheffer et al., 2001) and are indeed present in biogeochemical cycles, e.g. nutrient recycling (Richon and Tagliabue, 2021; Stone and Berman, 1993), light-phytoplankton feedback (Manizza et al., 2008) or temperature-phytoplankton

feedback (Károlyi et al., 2020). The finding that marine biogeochemical models can exhibit hysteresis, which is to be expected in nature, is of crucial importance as these models are used with an operational scope. The management and conservation of ecosystems requires special measures when they can exhibit hysteresis, as this limits recovery actions (Carpenter et al., 1999; Gladstone-Gallagher et al., 2024; Guarini and Coston-Guarini, 2024; Van De Leemput et al., 2016) and link them with different probabilities of (un)desired regime shifts (Stecher and Baumgärtner, 2022). In this case the role of phosphate is similar to what

found in lakes (Scheffer and Carpenter, 2003) and in facts lead to hysteresis.

    The absence of hysteresis in EXP-INI may indicate that differences in the initial conditions of the plankton species are not sufficient to cause the system to visit alternative dynamic regimes for the same environmental conditions. The difference between the results of EXP-SEQ and EXP-INI may be caused on how the simulations are initialized. In EXP-SEQ the initial conditions of a simulation has been provided from the end of the previous in the sequence of simulations, e.g. the initial

conditions of the forward simulation performed under a mean wind forcing of $5\,m/s$ are the last timestep of the simulation performed under a mean wind forcing of $4\,m/s$, so that each simulation has different initial conditions for all variables. In EXP-INI only the initial conditions related to plankton type concentrations are perturbed. The existence of alternative dynamic regimes under different environmental forcings but not for the same forcing, perturbing plankton initial conditions, is consistent with the findings that the long-term predictions of marine biogeochemical models are more strongly influenced by physics than

by the initialization of biogeochemical variables (Fransner et al., 2020).

    The experiment in which sequential simulations were performed (EXP-SEQ) has shown that the dynamic regimes can return to their original value when the perturbation of the environmental conditions is removed. Therefore, the modeled ecosystem exhibits reversible dynamic regimes under perturbation of the environment. The increase in SPM and air temperature led to a reversible response in all target indicators studied (see supplementary materials, Figures S3, S4, S7 and S8). The increase

in wind (see supplementary materials, Figures S1 and S2) led to a hysteretic response in some indicators (e.g. chlorophyll





bloom intensity or total annual chlorophyll production), but to a reversible response in others (e.g. species composition in chlorophyll bloom or intensity of DCM). Looking at the entirety of the simulations, i.e. each target indicator and each intensity of environmental forcings, the reversible response is more frequent than hysteresis. In contrast to simple models (e.g. the Hill model in Eq.1), complex dynamics can obscure the transition to alternative regimes and hysteresis (Dai et al., 2012), e.g.

spatial heterogeneity can influence the strength of hysteresis (Van Nes and Scheffer, 2005). Future studies could investigate whether the dominance of hysteresis or reversibility in marine biogeochemical models is influenced by the parameterization of the coupled physical model. Reversible responses to environmental conditions have been observed in marine ecosystems (e.g. for plankton (Molinero et al., 2013; Soulié et al., 2022) and fish (Sguotti et al., 2022)), terrestrial ecosystems (e.g. subalpine meadows (Ma et al., 2019) and grasslands (Bagchi et al., 2017)), and in physical states (e.g. Arctic sea ice (Armour et al.,

2011)). Since reversible systems can return to their original state, the study of reversibility is of great importance in ecological and conservation research and contributes to the dissemination of knowledge for policy change, climate change mitigation and ecological restoration (Buhr et al., 2024).

Ocean biogeochemical models operate in an inherently random and uncertain marine environment, as most natural phenomena do not follow strictly deterministic laws (Beddington and May, 1977). A stochastic formulation can therefore improve their

accuracy or reveal new dynamics (Occhipinti et al., 2024). For example, the birth and death of each individual in a population is a discrete and probabilistic event (Lindo et al., 2023; Melbourne, 2012) that causes random fluctuations in population size and is referred to as demographic stochasticity. Demographic stochasticity can lead to alternative regimes (DeMalach et al., 2021) and increase the realism of a population ecological model (Kaitala et al., 2006). While in the absence of noise we found that alternative regimes occur commonly for different environmental conditions, in the presence of noise (EXP-DEM) they

commonly occur for the same environmental condition, i.e. they are less affected by the change in environmental forcings (see Fig.5). We studied the effects of demographic stochasticity in the context of an increase in wind velocity, which induces an increased mixing in the water column. We may hypothesize that noise has a similar effect on mixing of plankton biomass in the water column, e.g. in the near-equilibrium limit Eq.3 resembles the Ornstein-Uhlenbeck process (Occhipinti et al., 2024), which describes a particle moving under Brownian motion with friction (Uhlenbeck and Ornstein, 1930). Multiplicative Noise may

overcome wind-driven mixing because it acts along the entire water column and directly affects plankton biomass. We found that when hysteresis occurs in the deterministic model, noise can increase or decrease the difference between the alternative dynamic regimes (see Fig.5, first and last columns). Furthermore, there is a non-monotonic relationship between the increase in the difference between the regimes and the noise intensity, i.e. an increase in noise intensity can cause both an increase and a decrease in this difference. This is the first time, up to our knowledge, that a non-monotonic relationship between hysteresis and

noise has been found, which could lead to the hypothesis of non-trivial noise induced phenomenon. Similar phenomena have been observed in other contexts, e.g. a non-monotonic relationship between the intensity of environmental stochasticity and the coexistence between plankton species and stochastic resonance was found in a similar model (Occhipinti et al., 2024). The fact that the system responds to the variation of noise intensity similarly to the variation of wind intensity (visiting two alternative dynamic regimes) allows for the hypothesis that demographic stochasticity is not required in the considered biogeochemical



model to visit different alternative dynamic regimes. On the other hand, the non-linear response of hysteresis suggests that some stronger phenomena can be sought in the presence of specific noise intensities.

Biogeochemical models use simplified schemes to describe numerous and diverse complex biological and chemical processes using a large number of parameters (e.g. the BFM has 200 parameters). These parameters are poorly constrained by theory and observations, which is why a sensitivity analysis of their impact on the results is recommended (Ciavatta et al., 2025; Mamnun et al., 2023; Prieur et al., 2019). Additionally, anthropogenic climate change leads to physical and biogeochemical changes in the ocean that affect the uncertainty of model predictions (Brett et al., 2021; Cao et al., 2014; Kwiatkowski et al., 2020). Our sensitivity analysis (EXP-BGC) has shown that aggregated ecosystem indicators (annual production and cycling) are less sensitive to parameter perturbation than seasonal dynamic indicators (chlorophyll bloom and DCM). Similarly, a lower sensitivity of aggregated ecosystem indicators, e.g. global primary production (Kriest et al., 2012), has already been observed. The sensitivity of the dynamic regimes is not increasing or decreasing with the perturbation of the environmental forcing (see Fig.6). This consistent accuracy, even under extreme environments, makes the BFM a valuable model for studying the effects of climate change (e.g. in the Mediterranean Sea (Reale et al., 2022)), which is particularly important as extreme events are increasing and may put at risk the resilience of marine ecosystems (Gruber et al., 2021). We have studied the effects of a constant alteration in the environmental conditions through numerical simulations, but a sudden change in the environment may cause other effects. Future studies could investigate the role of sudden extreme events, e.g. heat waves can trigger regime shifts (Soulié et al., 2022).

## 5   Conclusions

Alternative dynamic regimes has been found in the biogeochemical flux model (BFM), under the perturbation of its environmental forcings, consistently with field observations of regime shifts in plankton populations. The dynamic regimes generally return to their original state when the perturbation is removed, suggesting that the BFM describes a reversible system. Under certain environmental conditions, e.g. phosphate deficiency and increased wind velocity, hysteresis occurs in the dynamic regimes. The hysteresis is related to alternative regimes in the biogeochemical processes as well as in the trophic composition of the plankton community. However, we found that the perturbation of the initial conditions of the plankton alone is not sufficient to produce the observed hysteresis. This leads us to hypothesize that the environmental conditions are the main driver of the alternative dynamic regimes.

The introduction of noise into the biogeochemical processes, as demographic stochasticity, reduced the sensitivity of the dynamic regimes to the perturbation of environmental forcings and appeared to be the main driver of alternative regimes. Furthermore, the sensitivity analysis of the BFM parameters showed a consistent accuracy of the model even under extreme environmental forcings.

The ability to predict alternative dynamic regimes and the consistent accuracy argue for the use of modern biogeochemical models in predicting the state of the ocean. This is particularly important as hysteresis poses a challenge to management and conservation, as climate changes, perturbing the marine environment, cause extreme events.



*Code and data availability.* The code for running simulations of the BFM in the 1-D water column configuration are available at https://github.com/inogs/seamless-notebooks. The setup files to lunch a simulation with unperturbed forcings are available at https://doi.org/10.
5281/zenodo.15622626. The code with the implementation of demographic stochasticity in the BFM (Sect.2.3.3) can be found at https://github.com/inogs/bfmforfabm/tree/DemNoise.

*Author contributions.* CRediT: Guido Occhipinti: Conceptualization, Formal Analysis, Investigation, Methodology, Software, Validation, Visualization, Writing – original draft; Davide Valenti: Conceptualization, Methodology, Validation, Writing – review & editing; Paolo Lazzari: Conceptualization, Funding acquisition, Methodology, Project administration, Validation, Writing – review & editing

*Competing interests.* The authors declare that they have no conflict of interest.

*Acknowledgements.* The authors thank Vasilis Dakos for insightful discussions on alternative dynamic regimes. This work was supported by the National Biodiversity Future Center NFBC project: National Recovery and Resilience Plan (NRRP), Mission 4 Component 2 Investment 1.4 - Call for tender No. 3138 of 16 December 2021, rectified by Decree n.3175 of 18 December 2021 of Italian Ministry of University and Research funded by the European Union – NextGenerationEU. Davide Valenti acknowledges support from European Union - Next 435   Generation EU through project THENCE - Partenariato Esteso NQSTI - PE00000023 - Spoke 2 and from PRIN 2020 - DAMATIRA: aDvanced Analysis and Modeling of AcousTIc Responses of plAnts - B77G22000050001.



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
