# Peer review of "Alternative dynamic regimes of plankton communities in perturbed environments"

_EGUsphere, 2025_

## Referee Comment (RC1)

**Comments to the authors**

**General comments:**

The authors investigated the response of a biogeochemical models to extreme environmental changes. Specifically, they studied the response of the "system" to various perturbations and whether it was reversible or not. This article examines an interesting and important topic, i.e. how biogeochemical models and the biological systems they represent are responding to extreme perturbations, before and after it has occurred. However, I struggled to understand the analyses that have been done as well as the results and the conclusions. Note that my background is not in biogeochemical modelling, so I may not be the best person to review this work, which may explain my misunderstanding. I have tried to make some comments that should help less specialised readers to better understand. My aim is not to discourage the authors but rather trying to make the article clearer.

First, I think the authors need to clarify the vocabulary they use. For example, I do not understand what they mean by "*regime*", "*dynamic regime*", "*alternative regime*", "*system*", "*alternative dynamic regime*", "*operational model*", etc. The paragraph lines 103-109 page 4 also really confused me. For example, you say that you "*call the stable states dynamic regimes*" while to me, "stable" and "dynamic" are contradictory.

Second, the main figures are hard to understand. What are the authors trying to display? As you are talking about plankton dynamics, regime shifts, environmental data covering a given time period, etc, I was expecting to see time series, starting from a data t and ending at a time t+45 years (for example, a 45 years simulation is mentioned line 148 page 8), with maybe an indication that a perturbation was implemented at some point, so we can see the changes in the modelled plankton community, with and without the perturbation. Instead of that, we have a series of experiments, which make sense in the method, but no more in the results. For example, they say, for the EXP-SEQ experiment, that the regime returns to its original values, but when I am looking at Figure 3, I cannot really understand where I am supposed to see that. Furthermore, as the authors worked with a biogeochemical model, I was also expected to see how those modelled regime shifts impact the carbon cycle (for example) but only chlorophyll was investigated.

Third, I have been disappointed by the title, which mentions "plankton communities", and the method, which describes various plankton functional types (PFT). Therefore, I was expecting to see the changes observed for those different PFT (diatoms, dinoflagellates, microzooplankton, etc) in response to the environmental perturbations. Instead, the only variables describing plankton dynamic are chlorophyll bloom, deep chlorophyll maximum and total production of chlorophyll. I also wonder how is it possible to talk about dynamic regime of plankton community while changes in community composition

are not even displayed in any of the main figures? In the same way, the authors mention the seasonal cycles many times, but I cannot see how their results relate to the study of seasonality/phenology (except Fig S1?).

I am sorry for those comments. I think the study is of interest, but I have not been able to understand the results. I suggest the authors should rewrite their article, e.g. reconsidering how their results are presented as well as clarifying the vocabulary they use. It will not be a huge work. For example, they should not mention "*plankton community*" but directly "chlorophyll" in the title and throughout the text. I would also highlight that the model is applied at a fixed point in the Mediterranean Sea (if not in the title, at least in the abstract), so those results may not be applicable at a world scale(?). For the figure, maybe the authors should add an arrow below the panels, to highlight that the value of that environmental parameter is increasing forward and then backward, and also highlight where (and when?) the extreme values are reached for each environmental forcing. Adding a figure legend describing the meaning of the different lines/triangles would also help to understand quickly the figures after a first look.

---

## Referee Comment (RC2)

This manuscript presents a systematic numerical investigation of alternative dynamic regimes in a marine biogeochemical model. Overall, the experimental design is well-considered, and the exploration of hysteresis under extreme forcing is conceptually valuable. However, I have several major concerns regarding the experimental setup's realism, the manuscript's accessibility, and the interpretation of its findings. These issues currently limit the study's impact and the generalizability of its conclusions.

1. The core experimental design applies constant, extreme environmental forcings over 45-year periods (and up to 100 years in confirmatory runs) to reach a new steady state. While this is a standard numerical technique for bifurcation analysis, its connection to real ocean dynamics is not straightforward.

1) The real ocean is characterized by stochastic variability, episodic extreme events, and trends superimposed on seasonal cycles, not by decades of static, extreme conditions. Consequently, it is not straightforward to interpret what the discovered "regimes" and hysteresis loops mean for real plankton communities experiencing climate change. To me, these experiments can be interpreted as revealing potential system properties and thresholds. I would encourage the authors to make the interpretation and the ecological relevance of their experiments clearer in the manuscript.

2) Another issue is the physical plausibility of running a 1D water column model for 45-100 years under constant or altered forcing. A key limitation of 1D models is the lack of horizontal advection, which in the real ocean is essential for maintaining heat and salt budgets. Under constant surface forcing, a 1D model is prone to unphysical drift in temperature and salinity due to the absence of compensating horizontal fluxes (e.g., oceanic heat transport, lateral freshwater inputs). Did such a drift occur in your simulations? If so, how was it controlled or accounted for (e.g., through strong relaxation to observed profiles)? If not, please explain the physical rationale for the stability of the water column over such long timescales in this model setup.

3) Some of the applied perturbations (e.g., wind increase of +15 m/s, temperature increase of 15 degree, phosphate reduction to near-zero) appear exceptionally strong. Please comment on whether these ranges are physically plausible or should be viewed as purely theoretical limits.

2. The manuscript assumes considerable prior familiarity with the 1D model configuration and the BOUSSOLE site, which hinders comprehension.

1) A brief overview of the BOUSSOLE site is helpful (e.g., depth, typical seasonal stratification dynamics, trophic status) would help readers understand why the system responds to perturbations in particular ways (e.g., sensitivity to light vs. nutrients).

2) A concise summary (preferably with a figure or a clear paragraph) of the model's performance under unperturbed, climatological forcing is missing. Showing the modeled seasonal cycles of key variables (e.g., mixed layer depth, surface nutrients, chlorophyll) would establish a critical baseline and make the perturbed experiment results much easier to interpret.

3) The selection of specific forcings (especially phosphate and oxygen nudging), model parameters (10 were chosen), and target indicators (e.g., oxygen production, N remineralization) appears arbitrary without clear justification. Please provide a rationale for these choices early in the Methods. E.g., why are phosphate and oxygen selected for nudging? How were the 10 biogeochemical parameters chosen? What is the ecological significance of the selected target indicators?

4) Consider moving Section 2.4 ("Nudging of biogeochemical variables") before Section 2.3 ("Experiments setup"). This would help readers understand the experimental design from the outset.

3. The study draws broad conclusions about "marine biogeochemical models", yet all experiments, despite the perturbations of key physical forcings (wind, temperature), are conducted within the specific physical and ecological context of a single 1D location in the Mediterranean. While the applied perturbations are theoretically general, the baseline state of the system (e.g., its characteristic depth, stratification regime, initial nutrient levels, and plankton community) will strongly modulate the response. Therefore, the representativeness of the specific hysteresis thresholds and reversible behaviors found here for other regions (e.g., high-latitude seas or coastal upwelling zones with fundamentally different physical and biogeochemical dynamics) remains uncertain. The Discussion should explicitly address this limitation and speculate on how different baseline physical regimes might alter the propensity for, and manifestation of, hysteresis.

4. The sensitivity analysis (EXP-BGC) perturbs parameters within a fixed model structure. It is widely recognized that structural differences between models (e.g., food-web complexity, representations of remineralization) often cause greater divergence in projections than parameter variations alone. Therefore, the finding here should be softened to state that the model's response is robust to parameter uncertainty within this specific model structure. A discussion on how missing or oversimplified processes might influence the results is warranted.

5. Some organizational and presentational issues

1) The title emphasizes "plankton communities", but the results focus predominantly on biogeochemical flux indicators (chlorophyll peaks, DCM metrics, integrated production/remineralization). Shifts in plankton community composition (PFT biomass ratios) are mentioned but are not the primary lens of analysis. The title, abstract, and discussion should be aligned to accurately reflect the paper's focus, which is currently more on biogeochemical function regimes than on detailed community ecology.

2) Figure 2 is helpful, but the textual descriptions in Section 2.3 remain difficult to follow. Each experiment should be introduced with a clearer, plain-language statement of for its objective before delving into technical details.

3) The Discussion section currently reads somewhat like an extended interpretation of Results. While relevant literature is mentioned, the connections could be made more explicit and systematic. I suggest streamlining the discussion to better synthesize the core take-home messages from all experiments into a coherent narrative, explicitly compare and contrast the findings with the existing literature on regime shifts and hysteresis, and more clearly explore the implications of the work (e.g., for model development or for detecting early warnings) alongside its limitations and future directions.

Minor comments

L12-13: this sentence needs to be rephrased. It might lead readers to infer that the specific mechanisms simulated under the model's idealized setup have been directly validated by corresponding field measurements.

L48: "whether"

L85: "nitrate, phosphate, and silicate"

L86: should it be nitrate instead of ammonium?

L105: hard to follow why "stable states" are referred to as "dynamic regimes"

L106–109: hard to follow. Please rephrase.

L120: X instead of x?

L127: The caption of Fig. 1 indicates p = 18. Could you clarify whether the term "higher p" in the text refers specifically to values like 18, or whether hysteresis would also occur for moderately elevated values such as p = 2 or 3?

L140: delete the extra "a"

L150: Please clarify why the perturbation to species concentrations is applied only at the start of the second (backward) set of simulations, and not in the first (forward) set.

L151-152: The description of the perturbation to intracellular quotas is difficult to follow, primarily due to a lack of context about how these quota elements are defined and function within the model. A brief explanatory note would be helpful.

L214-215: fix the grammar

L221: where are these results presented?

L317–318: Please specify the primary climate-driven process postulated to increase SPM. Please also acknowledge contrasting drivers, such as anthropogenic reductions in sediment loads due to dam construction, which can decrease SPM and increase light availability.

L343: caused on -> caused by

L409: consistently -> consistent

Figure 1 (right panel) and its subsequent appearances (Figs. 3 & 4) include a gray line labeled as the "unstable dynamic regimes" of the Hill model. However, the main text provides no explanation of what an "unstable dynamic regime" represents in this context, how it is numerically identified or achieved in the model, or what its ecological interpretation might be. This omission makes the figure difficult to interpret. More importantly, while the theoretical Hill model plot includes these unstable branches, none of the actual numerical experiments with the BFM show or discuss analogous unstable states. Please consider clarifying in the text or simplifying the figure.

Figures 3 & 4: The bottom right panel is labeled "Annual Production", but its Y-axis is labeled "Total N remin. (0–100m) [mmol N/m2]". Please correct the label to accurately reflect the variable being plotted.

Figure 4: The caption states that "Hysteresis occurs only for a strong decrease in the phosphate nudging and is highlighted by a red circle." However, the two data points (triangles) enclosed within the red circle appear very close in value. Please clarify the quantitative criterion used to define this separation as a hysteresis loop.

Figure 5: In both the first and fourth columns of panels, a text box containing the number "0" is present. Please explain the meaning of this annotation in the figure caption.

Figure 6: The Y-axis label for the CV appears to include units. Isn't CV a dimensionless ratio?

Table 1 caption: annual mean instead of mean?

Table 2: In multiple rows, "July August" is listed without punctuation or conjunction.

Supplement: several text paragraphs currently placed at the end of Fig. S11 appear to be part of the general model description.

---

## Author Comment (AC1)

**Reply to Anonymous Referee #1**

We would like to thank the referee for its suggestions to improve the clarity of the manuscript and open it for a broader audience.

In this reply we give an overview of the main changes we did in order to address the referee's comments. Changes in the reviewed manuscript are highlighted in blue. Below you will find a point-by-point response to the comments, together with a description of the changes made to the manuscript. To address the comments of Reviewers 1 and 2, we split Figure 3 into two separate figures, one illustrating reversible responses and one illustrating hysteresis, and expanded them to include additional indicators related to plankton community structure and the carbon cycle. In addition, we introduced a quantitative indicator of hysteresis based on the coefficient of variation between alternative states, which also identifies a reversible response under wind forcing.

**RC1:** *First, I think the authors need to clarify the vocabulary they use. For example, I do not understand what they mean by "regime", "dynamic regime", "alternative regime", "system", "alternative dynamic regime", "operational model", etc. The paragraph lines 103-109 page 4 also really confused me. For example, you say that you "call the stable states dynamic regimes" while to me, "stable" and "dynamic" are contradictory.*

**Reply:** Thank you for pointing out that several key terms were not clearly defined in the manuscript. We agree that precise terminology is essential for the reader's understanding, and we have now added explicit and more detailed definitions of all specialized terms (e.g., *regime*, *dynamic regime*, *alternative regime*, *system*, *operational model*). We also clarified the paragraph in lines 103–109 to avoid confusion, particularly regarding the use of the term *dynamic regime* in relation to *stable states*.

**RC1:** *Second, the main figures are hard to understand. What are the authors trying to display? As you are talking about plankton dynamics, regime shifts, environmental data covering a given time period, etc, I was expecting to see time series, starting from a data t and ending at a time t+45 years (for example, a 45 years simulation is mentioned line 148 page 8), with maybe an indication that a perturbation was implemented at some point, so we can see the changes in the modelled plankton community, with and without the perturbation. Instead of that, we have a series of experiments, which make sense in the method, but no more in the results. For example, they say, for the EXP-SEQ experiment, that the regime returns to its original values, but when I am looking at Figure 3, I cannot really understand where I am supposed to see that. Furthermore, as the authors worked with a biogeochemical model, I was also expected to see how those modelled regime shifts impact the carbon cycle (for example) but only chlorophyll was investigated. [...] For the figure, maybe the authors should add an arrow below the panels, to highlight that the value of that environmental parameter is increasing forward and then backward, and also highlight where (and when?) the extreme values are reached for each environmental forcing. Adding a figure legend describing the meaning of the different lines/triangles would also help to understand quickly the figures after a first look.*

**Reply:** Thank you for pointing out that the figures are difficult to interpret. Because the perturbations in our experiments are constant in time and applied throughout each simulation, there is no clear 'before–after' transition that could be shown with time series. Our figures therefore focus on the long-term equilibrated behaviour after model transients

have decayed, and we now clarify this more explicitly in the manuscript. To improve readability, we will revise the figures to indicate more clearly the direction of the forward and backward forcing sequences. We will also add figures illustrating how the identified regime shifts affect carbon-cycle variables, not only chlorophyll. We trust these changes will make the results substantially easier to follow.

**RC1:** *Third, I have been disappointed by the title, which mentions "plankton communities", and the method, which describes various plankton functional types (PFT). Therefore, I was expecting to see the changes observed for those different PFT (diatoms, dinoflagellates, microzooplankton, etc) in response to the environmental perturbations. Instead, the only variables describing plankton dynamic are chlorophyll bloom, deep chlorophyll maximum and total production of chlorophyll. I also wonder how is it possible to talk about dynamic regime of plankton community while changes in community composition are not even displayed in any of the main figures? In the same way, the authors mention the seasonal cycles many times, but I cannot see how their results relate to the study of seasonality/phenology (except Fig S1?).*

**Reply:** Thank you for highlighting that our focus on aggregated chlorophyll metrics made the responses of individual plankton functional types (PFTs) difficult to appreciate. While detailed PFT-specific results were included in the Supplementary Information, we agree that their absence from the main figures obscured important aspects of community-level changes. We have therefore moved several key PFT-level figures into the main text to more clearly illustrate how community composition responds to environmental perturbations. We also clarify in the manuscript that the seasonal dynamics we refer to, namely the chlorophyll spring bloom and the deep chlorophyll maximum, are used as descriptors of the system's seasonal behaviour.

**RC1:** *I am sorry for those comments. I think the study is of interest, but I have not been able to understand the results. I suggest the authors should rewrite their article, e.g. reconsidering how their results are presented as well as clarifying the vocabulary they use. It will not be a huge work. For example, they should not mention "plankton community" but directly "chlorophyll" in the title and throughout the text. I would also highlight that the model is applied at a fixed point in the Mediterranean Sea (if not in the title, at least in the abstract), so those results may not be applicable at a world scale(?).*

**Reply:** Thank you for these final comments. We appreciate your perspective, and we agree that the manuscript will benefit from clearer terminology and a more transparent presentation of the results. To avoid implying a broader community-level analysis than is presented, we have revised the title to reflect more accurately that the study addresses biogeochemical dynamics, not exclusively plankton community composition. We also made explicit in both the abstract and the main text that the model is applied in a one-dimensional water-column configuration at a fixed location in the Mediterranean Sea, and therefore the findings should not be interpreted as globally representative.